# Zhile Capsule Exerts Antidepressant-Like Effects through Upregulation of the BDNF Signaling Pathway and Neuroprotection

**DOI:** 10.3390/ijms20010195

**Published:** 2019-01-08

**Authors:** Jiawei Wu, Tingting Zhang, Luping Yu, Shuai Huang, Yu Yang, Suyun Yu, Jun Li, Yuzhu Cao, Zhonghong Wei, Xiaoman Li, Yuanyuan Wu, Wenxing Chen, Aiyun Wang, Yin Lu

**Affiliations:** 1Jiangsu Key Laboratory for Pharmacolgy and Safety Evaluation of Chinese Materia Medica, School of Pharmacy, Nanjing University of Chinese Medicine, Nanjing 210023, China; wjwwjwwujiawei@163.com (J.W.); cassiett095@163.com (T.Z.); 20161329@njucm.edu.cn (S.H.); 20161333@njucm.edu.cn (Y.Y.); 15295511992@163.com (S.Y.); caoyuzhu0922@126.com (Y.C.); adonis_1978@126.com (Z.W.); xiaoman1205@163.com (X.L.); ywu@njucm.edu.cn (Y.W.); chenwx@njucm.edu.cn (W.C.); 2Zhejiang CN Strong Pharmaceutical Co., Ltd., Hangzhou 310000, China; paul@cnstrong.cn (L.Y.); lijun1@cnstrong.cn (J.L.); 3State Key Laboratory Cultivation Base for Traditional Chinese Medicine (TCM) Quality and Efficacy, Nanjing University of Chinese Medicine, Nanjing University of Chinese Medicine, Nanjing 210023, China; 4Jiangsu Collaborative Innovation Center of Traditional Chinese Medicine (TCM) Prevention and Treatment of Tumor, Nanjing University of Chinese Medicine, Nanjing 210023, China

**Keywords:** major depressive disorder, chronic unpredictable mild stress, systems pharmacology, brain-derived neurotrophic factor, neuroprotection

## Abstract

Major depressive disorder is now becoming a common disease in daily life, and most patients do not have satisfactory treatment outcomes. We herein evaluated the therapeutic effects of Zhile capsule and clarified the molecular mechanism. A rat model of chronic unpredictable mild stress-induced depression was established to assess the antidepressant-like effects of Zhile by using the sucrose preference test, open field test, forced swim test, tail suspension test and HPLC. Systems pharmacology was then performed to unravel the underlying mechanism which was confirmed by western blot, enzyme-linked immunosorbent assay, and qPCR. Zhile alleviated depression-like behaviors by upregulating the cAMP-CREB-BDNF (brain-derived neurotrophic factor) axis to exert neuroprotective effects. It may be beneficial to depressive patients in clinical practice.

## 1. Introduction

Currently, over 300 million people are suffering from major depressive disorder worldwide, as the leading cause for disability. Clinically, depression is characterized by dysthymic moods, loss of interest in things once pleasurable, anhedonia, feelings of despair, and loss of motivation. The complexity of this disease is caused by the interplay between multiple inherited genes and exposure to environmental stressors throughout life. Therefore, unravelling its neurological basis remains one of the foremost challenges in modern psychiatry [1,2].

Improving the treatment of depression urgently requires a suitable animal model. To date, animal models of chronic unpredictable mild stress (CUMS) have been used to study depressive-like behaviors. The CUMS model, which effectively mimics human life stressors, is probably the most investigated animal model of depression. The first CUMS paradigm was introduced by Katz et al., consisting of a three-week protocol during which rats were exposed to several transient and repetitive mild stressors such as unpredictable shocks, food deprivation, reversal of day/night cycle, and shaker stress [3]. The rat model of CUMS resembles a number of behavioral and neuroendocrine alterations in depressive patients [4]. The clinical symptoms and signs of depression in rats, such as anhedonia (loss of interest or pleasure in events that usually would be enjoyed) and helplessness, are usually examined using the sucrose preference test (SPT) and forced swim test (FST), respectively.

Chronic stress can induce morphological and functional changes in different brain areas, such as the hippocampus and cerebral cortex, possibly leading to degenerative-like changes [5]. Accumulating evidence has verified that CUMS participated in the development of depression following various molecular and cellular mechanisms, and regulated the neurotransmitter content and the cAMP-CREB-BDNF (brain-derived neurotrophic factor) signaling pathway. Besides, altered BDNF expression may be related with the etiologies of depression.

Zhile capsule (ZL), a Traditional Chinese Medicine (TCM) formula-based capsule, is composed of 16 herbs, including *Schisandra chinensis*, *Semen Ziziphi Spinosae*, *Codonopsis Radix*, *Cinnamomi Ramulus*, *Rehmanniae Radix Praeparata*, *Rhizoma Dioscoreae*, *Cortex Moutan*, *Gardeniae Fructus*, *Sennae Folium*, *Inulae Flos*, *Caulis Polygoni Multiflori*, *Trichosanthes kirilowii*, *Bambusae Caulis* in *Taeniis*, *Lophatherum gracile*, *Ophiopogon japonicus*, and *Zingiberis rhizome*. Many drugs containing one of the above components have a variety of biological effects on the central nervous and immune systems, accompanied by antidepressant effects. However, the neuroprotective and anti-apoptotic effects of ZL on an animal model of CUMS-induced depression or the potential mechanisms are still unclear. Thereby motivated, we herein evaluated the therapeutic effects of ZL on a rat model of CUMS-induced depression, and explored the molecular mechanism from the perspectives of neuroprotection and apoptosis inhibition.

The rat model of CUMS was first established. The antidepressant-like effects of ZL were then confirmed by SPT, FST, tail suspension test (TST), and open field test (OFT). In addition, we used a systemic approach to study the possible mechanism due to the complexity and diversity of ingredients in TCM formula, identifying the main pathway of BDNF and apoptosis.

## 2. Results

### 2.1. ZL Attenuated Depressive-Like Behaviors of CUMS Rats

To verify whether ZL have antidepressant-like effects on CUMS rats, we conducted our experiments as Figure 1A. As shown in Figure 1B, the body weight of CUMS rats plummets from the third week, whereas ZL (0.735 g/kg, 2.205 g/kg, 6.615 g/kg) significantly increases the body weight (*p* < 0.01, *p* < 0.05, and *p* < 0.001, respectively) at the end of experiment. 10 mg/kg fluoxetine also had the same effects (*p* < 0.01). As an approach for evaluating the antidepressant effects of ZL, SPT was performed after five weeks of CUMS exposure. The increases in sucrose consumption suggested potent antidepressant-like effects of ZL on the CUMS-exposed rats. The saccharification indices of different groups were similar before test. Nevertheless, a five-week CUMS exposure significantly reduced the sucrose consumption of stressed rats compared with that of the control group (*p* < 0.01). After 35 days of experiment, fluoxetine markedly increased sucrose preference percentage (*p* < 0.05). ZL (0.735 g/kg, 2.205 g/kg, 6.615 g/kg) significantly increased the sucrose preference compared to that of CUMS model rats (*p* < 0.05, *p* < 0.05, *p* < 0.01, respectively). Figure 1D exhibits the effects of ZL on the immobility time in TST. ZL (2.205 g/kg, 6.615 g/kg) significantly decreased the duration of immobility (*p* < 0.05 and *p* < 0.01, respectively) compared with that of the CUMS model group. Also, ZL at 2.205 g/kg and 6.615 g/kg significantly shortened the immobility time in FST (*p* < 0.01 and *p* < 0.001, respectively) compared with that of the CUMS model group (Figure 1E). Fluoxetine also increased immobility times both in TST and FST (*p* < 0.01 and *p* < 0.01, respectively). We next used open-field test to further evaluate the antidepressant-like effects of ZL. Before CUMS procedure, all groups showed comparable crossing score and rearing score. At day 35, we found that fluoxetine-treated mice showed a significant increase in crossing score and rearing score (*p* < 0.01 and *p* < 0.001, respectively). Meanwhile, ZL (2.205 g/kg, 6.615 g/kg) increased the crossing score (*p* < 0.01 and *p* < 0.001, respectively) and ZL (0.735g/kg, 2.205 g/kg, 6.615 g/kg) increased the rearing score (*p* < 0.01, *p* < 0.001 and *p* < 0.001, respectively) (Figure 2A,B). Taken these results together, ZL and fluoxetine exhibited considerable antidepressant-like behavior induced by CUMS.

### 2.2. ZL Modulated Hippocampal Monoaminergic System Functions of CUMS Rats

In patients undergoing stress and depression, the concentration of three monoamine neurotransmitters—namely, norepinephrine (NE), 5-hydroxytryptamine (5-HT), and dopamine (DA)—significantly decreased. We then test the three monoamine neurotransmitters in hippocampus of rats using HPLC-ECD method. The concentrations of NE, 5-HT, and DA in the hippocampus are presented in Figure 2C–E. CUMS exposure reduced hippocampal NE, 5-HT, and DA levels. Compared to the CUMS group, fluoxetine treatment markedly increased DA, 5-HT and NE levels (*p* < 0.05, *p* < 0.001 and *p* < 0.001, respectively). ZL (0.735 g/kg, 6.615 g/kg) increased DA levels (*p* < 0.001). Furthermore, ZL (0.735 g/kg, 2.205 g/kg, 6.615 g/kg) raised 5-HT levels (*p* < 0.001, *p* < 0.01 and *p* < 0.001, respectively), and reversed the decrease of NE levels (*p* < 0.001, *p* < 0.01 and *p* < 0.01, respectively). Collectively, ZL modulated the functions of monoaminergic system in the hippocampus of CUMS rats by upregulating hippocampal NE, 5-HT, and DA levels.

### 2.3. Systems Pharmacology Revealed Possible Therapeutic Role of the BDNF Signaling Pathway in Depression

ZL comprises sixteen herbs, but the mechanism by which ZL treats depression remains unclear due to the complexity of ingredients in TCM herbs. Here, we performed a systemic approach to investigate the possible pharmacological mechanism of ZL. The workflow of study’s approach is presented in Figure 3.

A total of 1005 compounds from ZL were collected. To screen active compounds, we tested the ADME properties of the ingredients, including Oral bioavailability (OB), Drug-likeness (DL), Caco-2 permeability (Caco-2), Blood-brain barrier (BBB) and Drug half-life (HL) using the criteria of OB% ≥ 30%, DL ≥ 0.18, Caco-2 ≥ −0.4, BBB ≥ −0.3 and HL ≥ 4 [6]. At last, 63 potential compounds passed the ADME screening (Figure 3A). Additionally, 31 compounds with poor screening parameters, which had potential antidepressant-like effects according to previous literatures, were also selected as active compounds. These 94 compounds were considered as active compounds and their ADME parameters are listed in Appendix A.

TCM formulas may exert diverse physiopathological effects through multiple targets [7]. Therefore, we used a systemic approach to predict the potential targets of ZL (Figure 3B) [8]. A total of 287 potential targets were predicted: 40 for *S. chinensis*, 24 for *Semen Ziziphi Spinosae*, 148 for *Codonopsis Radix*, 84 for *Cinnamomi Ramulus*, 31 for *Rehmanniae Radix Praeparata*, 73 for *Rhizoma Dioscoreae*, 180 for *Cortex Moutan*, 134 for *Gardeniae Fructus*, 58 for *Sennae Folium*, 202 for *Inulae Flos*, 39 for *Caulis Polygoni Multiflori*, 10 for *T. kirilowii*, 37 for *O. japonicus*, and 58 for *Z. rhizoma*. Notably, some herbs had similar targets. For instance, most herbs may affect monoamine oxidases which are key enzymes associated with the metabolism of depression-related monoamines such as serotonin, norepinephrine, and DA [9]. The detailed target information is available in Appendix A.

We then constructed a compound-target network based on the active compounds and their potential targets to clarify the complex interactions between them. As shown in Figure 4, the network contains 403 nodes and 1846 compound-target interactions. The degrees of quercetin, beta-sitosterol, and luteolin in the whole network were 298, 156, and 115, acting on 138, 38, and 57 targets, respectively. In other words, they were crucial members in the network. Existing in various herbs, they may have synergistic effects. It has previously been reported that quercetin attenuated the brain oxidative stress and mediated the neuroinflammation-apoptotic cascade, thus exerting antidepressant-like effects [10,11]. Luteolin may influence BDNF and its downstream apoptotic proteins such as Bax and Bcl-2 to significantly combat depression [12]. Beta-sitosterol may resist depression by elevating the levels of neurotransmitters such as norepinephrine, 5-HT, and its metabolite 5-HIAA [13]. Therefore, these targets may be crucial members of ZL. The detailed compound-target information is listed in Appendix A.

A total of 354 genes were collected from five databases (OMIM, DrugBank, PharmGKB, GAD, and TTD) and identified as depression-related targets. Sixteen herbs of ZL shared 41 targets with those related with depression, demonstrating a potential relationship between ZL and depression. The detailed target information is available in Appendix A.

Instead of functioning independently, genes and proteins work on multiple levels via interconnected molecular networks and pathways [14]. Thus, we selected proteins as nodes to generate the network (Figure 3C). First, we constructed a protein–protein interaction (PPI) network of ZL-related targets obtained by previous screening and prediction (8283 nodes and 185,028 edges). Next, we constructed a PPI network of depression-related targets (5182 nodes and 125,234 edges). Then we merged these two networks, giving a core protein–protein interaction (CPPI) network consisting of 4113 nodes and 109,939 edges. To identify the essential proteins in this CPPI network, the cytoscape plugin CytoNCA was thereafter used for centrality analysis [15]. In a network, a node is referred to as hub if its degree exceeds twice the median degree of all nodes [16]. We screened the main hubs of the network by the following six topological features, i.e., ‘degree centrality (DC)’, ‘betweenness centrality (BC)’, ‘closeness centrality (CC)’, ‘eigenvector centrality (EC)’, ‘network centrality (NC)’, and ‘local average connectivity (LAC)’. The medians of ‘DC’, ‘BC’, ‘CC’, ‘EC’, ’NC’, and ’LAC’ were 107, 8567.066, 0.4497922, 0.0208708, 23.436096, and 16.107143, respectively. Finally, 243 candidate targets were we considered as the main hubs. The detailed topological features of the CPPI network and 243 candidate targets are shown in Appendix A.

The potential pathways involving the candidate targets were further studied by using the Cytoscape plugin ClueGO (Figure 3D) [17]. As shown in Figure 5, BDNF, cell cycle and AGE/RAGE are the most related signaling pathways. Notably, the BDNF pathway plays an important role in depression. BDNF is one of the mammalian neurotrophins that contribute to the survival, growth and maintenance of neurons. Growing evidence has implicated that BDNF was strongly associated with depression [18]. Neurogenesis in the adult brain is an essential form of neuroplasticity involved in learning, memory, and emotional regulation. Thus, the proliferation and survival of new neurons in the hippocampus are of great importance to depression. Through binding Trk receptors, mature BDNF can activate its downstream PI3K/AKT and MAPK signaling to facilitate the proliferation and survival of neurons. Finally, BDNF may exert neuroprotective effects by regulating apoptosis-related proteins such as Bcl-2, Bax, and caspase-3 [19]. Given that the candidate targets were mostly involved in the BDNF signaling pathway, ZL may combat depression by acting on this pathway. The detailed information of target pathways is shown in Appendix A.

### 2.4. ZL Exerted Antidepressant Effects by Regulating the cAMP-CREB-BDNF Signaling Pathway

As mentioned above, ZL may exert antidepressant-like effects via the BDNF signaling pathway. Since CREB and BDNF are important neurotrophins participating in the BDNF signaling pathway and affecting hippocampal neurogenesis, BDNF, CREB, and p-CREB protein levels were determined to test the underlying effects on the rat hippocampus (Figure 6A–C). Compared with the control group, the level of BDNF in the hippocampus of the CUMS group significantly decreased (*p* < 0.05), which was significantly reversed by 6.615 g/kg ZL administration (*p* < 0.01). Meanwhile, the p-CREB/CREB level significantly increased compared with that of the CUMS group after administration with 2.205 g/kg and 6.615 g/kg ZL (*p* < 0.01 and *p* < 0.001, respectively). Furthermore, the BDNF mRNA level of the high-dose ZL group (6.615 g/kg) was higher than that of the model group (Figure 6E). As a secondary messenger, cAMP can upregulate CREB phosphorylation. To further investigate the cAMP/CREB/BDNF pathway, we detected the cAMP levels in the hippocampus. As shown in Figure 6D, the cAMP level of the CUMS group significantly decreased compared with that of the control group (*p* < 0.05), but ZL significantly raised the level compared with that of the model group (*p* < 0.05). Thus, ZL affected cAMP level and promoted CREB phosphorylation, eventually upregulating BDNF protein and mRNA levels.

### 2.5. ZL Inhibited Apoptosis by Upregulating BDNF to Exert Neuroprotective Effects on CUMS Rats

To determine whether ZL elicited neuroprotective effects via the cAMP-CREB-BDNF pathway, we further detected BDNF downstream proteins related to apoptosis, such as Bcl-2, Bax, and cleaved caspase-3. Western blot showed increased levels of Bax and cleaved caspase-3 but decreased level of Bcl-2 in the hippocampus of CUMS rats compared to those of normal rats (Figure 7A–C), which were reversed by treatment with ZL. qPCR showed elevated level of Bcl-2 in the 6.615 g/kg ZL group compared to that of the CUMS group, although the Bax level remained unchanged (Figure 7D,E). Taken together, ZL was potentially neuroprotective through upregulation of BDNF and inhibition of apoptosis, validating the mechanism mentioned above.

## 3. Discussion

Nowadays, depression has become a widespread chronic disease which is related to the normal emotion of sadness, but it does not remit when the external cause for sadness emotion dissipates. It is typified by low mood, lack of energy, sadness, inability to enjoy life, and even urges to commit suicide. So far, depression is a heterogeneous disorder without established mechanisms. Depression may be related to genetics, monoamine deficiency, stress, altered glutamatergic neurotransmission, reduced GABAergic neurotransmission and so on. However, the treatment outcomes of most depressive patients are unsatisfactory, thereby requiring effective therapies.

Herein, ZL, a TCM formula comprising sixteen different herbs, exerted antidepressant-like effects on a rat CUMS model. It increased body weight and sucrose consumption, and shortened the immobility time in FST and TST. Surprisingly, ZL showed comparable antidepressant-like effect to fluoxetine, a drug already on the market.

Given the complexity and diversity of ingredients in TCM formula, it may be difficult to unravel the mechanism by which ZL treated depression, so the systemic approach was used. We first generated a pool of active compounds in ZL by ADME screening. Subsequently, we predicted the targets of ZL and constructed a compound-target network to gain insight into compound–target interactions. To further identify the key regulators of these targets which played important roles in treating depression, we generated a CPPI network based on two PPI networks, one for the targets of ZL and the other for depression-related targets. By screening the topological features of the CPPI network, we obtained the main proteins which may contribute to depression treatment. Using ClueGO, the BDNF signaling pathway was determined as the predominant one.

BDNF is a neurotrophic peptide which is critical for axonal growth, neuronal survival and synaptic plasticity, also as the link between stress and neurogenesis. Mature BDNF can activate its downstream PI3K/AKT and MAPK signaling, benefiting the proliferation and survival of new neurons. It is well-documented that BDNF level decreased in depressive patients or animal models, but increased after antidepressant therapies. As an active compound from *Codonopsis Radix*, apigenin has been reported to exhibit antidepressant-like effects through upregulation of BDNF level in the rat hippocampus. Moreover, both *Cortex Moutan* and *Cinnamomi Ramulus* contain eugenol, so they are also neuroprotective by upregulating the hippocampal BDNF level. Hence, we studied whether ZL affected the BDNF pathway in CUMS rats. After treatment with this formula, the BDNF protein and mRNA levels in the rat hippocampus significantly increased. Besides, cAMP level increased after ZL treatment. Also, it showed neuroprotective effects by inhibiting apoptosis. Thus, ZL may elicit antidepressant-like effects by activating the BDNF pathway and protecting neurons from apoptosis.

Regardless, this study still has some limitations. First, targets were predicted based on databases and structures, so not all potential targets have been found and other mechanisms may exist. Second, we did not take the proportion of each compound into consideration, so it may be imprecise to attribute the therapeutic effects of ZL to all compounds that passed the screening. Moreover, not all the components of ZL were discovered.

In short, ZL exerted remarkable antidepressant-like effects on CUMS via the cAMP-CREB-BDNF pathway, which may provide new prescription rules for TCM formulas used to clinically treat depression.

## 4. Materials and Methods

### 4.1. Reagents

ZL was gifted by Zhejiang CN Strong Pharmaceutical Co., Ltd. (Hangzhou, Zhejiang Province, China). Fluoxetine Hydrochloride (Lilly Suzhou pharmaceutical Co., Ltd., Suzhou, China, cat no. 6620A). HPLC-MS-grade 5-hydroxytryptamine (5-HT), 5-hydroxyindoleacetic acid (5-HIAA), dopamine (DA), homovanillic acid, and 3,4-dihydroxyphenylacetic acid were purchased from Sigma-Aldrich (St. Louis, MO, USA). Enzyme-linked immunosorbent assay (ELISA) kit for cAMP was bought from Jinyibai Biotechnology Co., Ltd., Nanjing, China (cat no. JEB-13658). Primary antibodies against BDNF, p-CREB, and CREB were purchased from Affinity Biosciences, lnc. (Cincinnati, OH, USA. cat no. DF6387, AF6188, AF3189, respectively). Primary antibodies against Bax, Bcl-2 and caspase-3 (cleaved caspase-3) were obtained from ABclonal (Wuhan, China. cat no. A12009, A0208, A2156, respectively). Primary antibody against GAPDH and horseradish peroxidase-conjugated anti-rabbit secondary antibodies were acquired from Bioworld (Nanjing, China. cat no. AP0063).

### 4.2. Animals

Male Sprague-Dawley rats (4–6 weeks of age, 180–210 g) were provided by Beijing Vital River Laboratory Animal Technology Co., Ltd. (Beijing, China). The animals were housed (three per cage) under standard laboratory conditions (room temperature: (22 ± 2) °C; humidity: (50 ± 5)%) with a light/dark cycle of 12/12 h (lighting on at 7:00 a.m.). All experimental protocols were approved by the Animal Care and Use Committee of Nanjing University of Chinese Medicine (Nanjing, China) and conducted conforming to the Guidelines for the Care and Use of Laboratory Animals (ACU-28(20161229), 29 December 2016).

### 4.3. CUMS Procedure

The CUMS procedure was slightly modified from that of Willner [20]. Specific details of the CUMS procedure were as follows: (1) food deprivation for 24 h; (2) water deprivation for 24 h; (3) horizontal vibration for 15 min; (4) cage tilting by 45°; (5) cage soiling; (6) swimming in 4 °C water for 5 min; (7) light/dark perversion; (8) body restriction for 2 h; (9) treatment at 45 °C for 5 min; (10) tail pinching at 1 cm from the tip. Each animal was exposed to one stress per day individually and randomly. This paradigm was devised to maximize unpredictability, in which the stressors were applied in the seemingly random SPT.

#### 4.3.1. SPT (Sucrose Preference Test)

SPT was conducted as reported previously with slight modifications [20]. Briefly, each animal was separately housed, and simultaneously given two bottles that were filled with 1% sucrose solution (w/v) and tap water, respectively. After 24 h, one bottle of sucrose solution was replaced by tap water for another 24 h. After adaptation, 24 h of water and food deprivation was performed. To prevent possible position preference, the distances from the two bottles to the animals were equal, and their positions were interchanged every 12 h. SPT was conducted in dark (7:00–9:00 p.m.), during which the animals were given two bottles filled with sucrose solution (1%, w/v) and tap water, respectively. Fluid consumption was monitored for 4 h, and then the bottles were removed and weighed. The baseline SPT was performed before stress and during CUMS weekly. Sucrose preference was calculated by the following formula: sucrose preference = (sucrose intake (g))/(sucrose intake (g) + water intake (g)) × 100%.

#### 4.3.2. TST (Tail Suspension Test)

TST was performed according to previous studies [21]. Briefly, on the 36th day, the animals were fixed on a tail suspension monitor at 1 cm from the tail tip in a suspended state, and the head was over 15 cm from the ground. After 2 min of adaption, the immobility time was recorded for 4 min. The animals were considered immobile only when they were hanging passively and completely motionless. Immobility was defined when the animals stopped struggling and the body was vertically suspended. Technical observers were blinded to the whole experiment.

#### 4.3.3. FST (Forced Swim Test)

FST was performed according to previous studies [22]. Briefly, on the 36th day, each animal was placed in an organic glass drum (height: 50 cm; diameter: 20 cm) filled with water (depth: 35 cm; temperature: (25 ± 1) °C) for 6 min, and the time of immobility during the last 4 min was detected. The animals were considered immobile when they were floating motionless or only slightly moving to keep their heads above water. Water was refreshed following each test. The technical observers were blinded to the whole experiment.

#### 4.3.4. OFT (Open Field Test)

The locomotor activities of rats were measured by OFT [21]. The apparatus floor (100 × 100 × 42 cm) had a black line equally divided into 25 squares. The floor and walls of the apparatus were black. The animals were individually placed in the middle of the box under dim light conditions using a ceiling light with a red bulb (50 W). The locomotion activities were evaluated by recording the number of grid crossing (three claws into the grid was counted once) and that of vertical standing for 5 min. After each test, the floor was wiped thoroughly with 75% ethanol solution to remove possible clues from the previous animal. The technical observers were blinded to the whole experiment.

### 4.4. Collection of Compounds in ZL

The compounds of each herb in ZL were collected from TCMSP (http://lsp.nwu.edu.cn/tcmsp.php), TCMID (http://www.megabionet.org/tcmid/), and TCM Database@Taiwan (http://tcm.cmu.edu.tw/) for further screening.

#### 4.4.1. Screening of Parameters of Active Compounds

OB: Oral bioavailability represents the percentage of an orally administered dose of unchanged drug that reaches the systemic circulation, which reveals the convergence of the absorption, distribution, metabolism, and excretion (ADME) process. High OB is often a key index for determining the drug-like properties of bioactive molecules as therapeutic agents.DL: Drug-likeness is a qualitative concept used in drug design to estimate how “drug-like” a prospective compound is, which helps to optimize pharmacokinetic and pharmaceutical properties, such as solubility and chemical stability.Caco-2: Caco-2 permeability is often used to represent the intestinal epithelial permeability, indicating the passive diffusion ability of drugs across the intestinal epithelium.BBB: The blood–brain barrier is anatomically characterized by the presence of intercellular tight junctions between continuous non-fenestrated endothelial cells, limiting the passage of proteins and potentially diagnostic and therapeutic agents into the brain parenchyma. It is critical for evaluating the penetration capacities of compounds into the central nervous system.HL: Drug half-life (t_1/2_), which is defined as “the time taken for the amount of compound in the body to fall by half”, may be the most important property, because it works as the timescale over which the compound may elicit therapeutic effects.

In this study, active compounds were screened by setting parameters as follows: OB ≥ 30%, DL ≥ 0.18, Caco-2 ≥ −0.4, BBB ≥ −0.3, and HL ≥ 4 [23].

#### 4.4.2. Drug Target Prediction for ZL and Construction of Compound–Target Network

To predict the potential targets of active compounds in ZL screened above, we used the in silico prediction model, chemogenomics methods, and databases [8]:
The in silico prediction model efficiently integrates chemical, genomic, and pharmacological information for drug targeting on a large scale, based on two powerful methods: random forest and support vector machine. When drug targets are identified, the proteins with an output expectation value (E-value) of support vector machine >0.7 or random forest >0.8 are potential targets.SEA search tool (http://sea.bkslab.org/), as an online search tool for the similarity ensemble approach, relates proteins based on the chemical similarities to their ligands. The final score is expressed as E-value indicating the structural similarity of each drug to the ligand of each target.TCMSP (http://lsp.nwu.edu.cn/tcmsp.php), as a noncommercial TCM database, highlights the role that the systems pharmacology plays in TCM discipline. Users can obtain the targets of compounds based on extensively studied pharmacology and clinical knowledge.

Thus, we combined these three methods and finally obtained the putative targets for ZL.

To gain insight into the relationships between compounds and targets, we constructed a compound-target network using Cytoscape software: https://cytoscape.org/ (Version 3.2.1).

#### 4.4.3. Identification of Depression-Related Targets

To further investigate the known depression-related targets, we searched five databases: Drugbank (http://www.drugbank.ca/, Version 5.1.1), OMIM (http://www.omim.org/, last updated: 30 June 2018), GAD (http://geneticassociationdb.nih.gov/, last updated: 1 September 2014), PharmGKB (https://www.pharmgkb.org/index.jsp, last updated: 7 April 2016) and TTD (http://database.idrb.cqu.edu.cn/TTD/, last updated: 15 September 2017). We searched these databases using key words “depression” and collected the depression-related targets in total.

#### 4.4.4. Construction of Protein–Protein Interaction (PPI) Network and Identification of Topological Features

An interaction network for the potential targets of ZL was constructed by using a Cytoscape plugin Bisogenet and visualized by Cytoscape software (Version 3.2.1) [24]. This plugin allows the searching of molecular interactions from well-known interaction databases including Database of Interacting Proteins, The Biological General Repository for Interaction Datasets, the Human Protein Reference Database, the Biomolecular Interaction Network Database, the Molecular INTeraction Database and the IntAct Molecular Interaction Database. We also constructed an interaction network for depression-related targets using the same method. Subsequently, these two networks were merged to obtain a core protein–protein interaction (CPPI) network.

Using a Cytoscape plugin CytoNCA [15], we performed network centrality analysis by calculating six topological features: ‘degree centrality (DC)’, ‘betweenness centrality (BC)’, ‘closeness centrality (CC)’, ‘eigenvector centrality (EC)’, ‘network centrality (NC)’, and ‘local average connectivity (LAC)’. The definitions and computational formulas of these parameters have previously been defined, representing the topological importance of a node in the network.

#### 4.4.5. Pathway Enrichment Analysis

We then used ClueGO [17], a Cytoscape plugin visualizing the non-redundant biological terms for large clusters of genes in a functionally grouped network, to enrich the key pathways involving candidate targets. A *p* value cut-off of ≤0.05 was considered significant. The ClueGO network was created by using kappa statistics, reflecting the relationships between the terms on the basis of the similarities between their associated genes.

### 4.5. Quantitative Real-Time PCR

At the end of the CUMS procedure, the rats were sacrificed by cervical dislocation. Then the hippocampus was rapidly removed from the brain on ice. After pretreatment, total RNA was extracted using a TRIzol extraction kit (Invitrogen, Carlsbad, CA, USA). Afterwards, cDNA was synthesized using a reverse transcription kit (Vazyme Biotech Co., Ltd, Nanjing, China). SYBR Green real-time PCR amplification and detection were performed using ABI 7500 system (Applied Biosystems, Foster City, CA, USA). The following primers were used: BDNF (forward: 5′ CGAGACCAAGTGCAATCCCA 3′; reverse: 5′ GTACGACTGGGTAGTTCGGC 3′). Bcl-2 (forward:5′ AAGCCGGGAGAACAGGGTAT 3′; reverse: 5′ CGCGGAGTCTTCATCTCCAG 3′). Bax (forward: 5′ AAGACAGGGGCCTTTTTGCTA 3′; reverse: 5′ TCCAAGGTCAGCTCAGGTGT). GAPDH (forward: 5′ AACTTTGGCATCGTGGAAGG 3′; reverse: 5′ GTGGATGCAGGGATGATGTTC 3′).

### 4.6. Western Blotting

Hippocampus tissue (approximately 50 mg) was homogenized in 0.5 mL of radioimmunoprecipitation assay buffer (Thermo Fisher Scientific, Waltham, MA, USA) for protein extraction and centrifuged at 12,000× *g* at 4 °C for 15 min. The supernatant was collected and protein concentrations were measured with a bicinchoninic acid protein assay kit (Thermo Fisher Scientific, Shanghai, China). Western blotting was performed as previously described [25]. Briefly, proteins were separated by sodium dodecyl sulphate-polyacrylamide gel electrophoresis and immunoblotted with antibodies against p-CREB, CREB, BDNF (1:1000; Abcam, San Francisco, CA, USA), and GAPDH (1:40,000; Bioworld, Visalia, CA, USA). The reaction was visualized by enhanced chemiluminescence reagent (Biosharp, Wuhan, China). Bio-Rad imaging system was used to quantify protein levels. All experiments were performed in triplicate.

### 4.7. HPLC-ECD

After five weeks of CUMS exposure, the brain was removed immediately, from which the hippocampus was quickly stripped on the ice tray, immediately frozen in liquid nitrogen and stored at −80 °C. The sample was premixed with perchloric acid solution (containing 0.1 M HClO_4_, 0.1 mM Na_2_EDTA, and 2 × 10^−7^ M DHBA, as described previously). After ultrasonic homogenization, the homogenate was centrifuged at 20,000 rpm (4 °C) for 30 min. Then the purified supernatant (10 μL) was subjected to HPLC on a reverse phase column: DIONEX Acclaim (R) rapid separation liquid chromatograph, 2.1 × 100 mm C18, 2.2 μm. The mobile phase consisting of 90 mM NaH_2_PO_4_, 50 mM citric acid, 1.7 mM OSA, 50 μM EDTA, and 5% acetonitrile was pumped at a flow rate of 0.2 mL/min. The compounds were detected electrochemically by a cell containing a glassy working carbon electrode with an applied oxidation potential of +0.70 V against an in situ Ag/AgCl reference electrode. The keep column temperature was 38 °C. External standard curves were used to quantify the amounts of monoaminergic transmitters including NE, DA, and 5-HT in each sample, using the areas under the curves.

### 4.8. cAMP ELISA

The concentration of cAMP in the lysate of hippocampal neurons in each group was measured using a rat cAMP ELISA kit according to the manufacturer’s instructions.

### 4.9. Statistical Analysis

All data were expressed as mean ± standard deviation (SD). Inter-group differences were compared using one-way analysis of variance, with the Dunnett’s post-hoc test for multiple testing. *p* < 0.05 was considered statistically significant. Statistical analysis was carried out using GraphPad Prism 7.0 software (https://www.graphpad.com/scientific-software/prism/).

## 5. Conclusions

In summary, ZL showed marked antidepressant-like and neuroprotective effects on a rat model of CUMS-induced depression, which may allow the effective treatment of major depressive disorder. As evidenced by network pharmacology, ZL was closely associated with the BDNF pathway. This method may help further understand the mechanisms of TCM drugs.

## Figures and Tables

**Figure 1 ijms-20-00195-f001:**
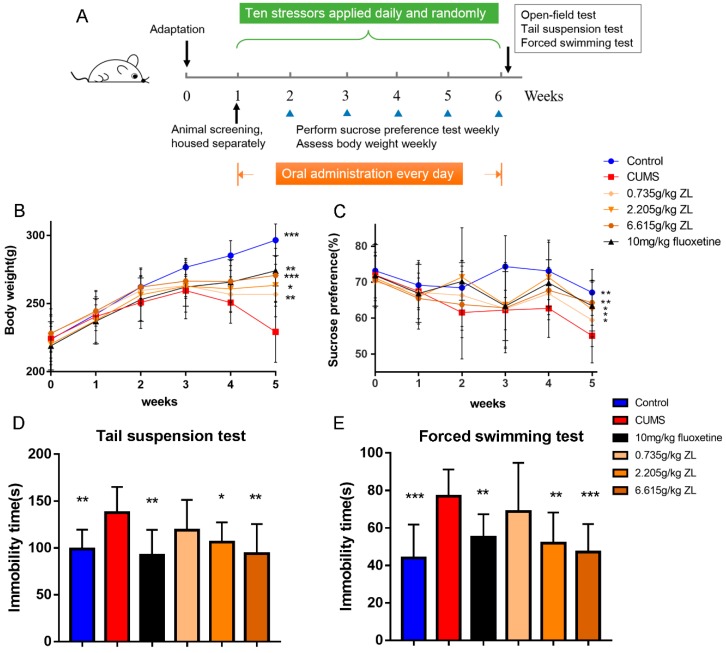
Effects of Zhile (ZL) on depressive-like behaviors of chronic unpredictable mild stress (CUMS) rats. CUMS procedure and experimental design (**A**). Effects of ZL on body weight (**B**), sucrose preference test (SPT) (**C**), tail suspension test (TST) (**D**), and forced swim test (FST) (**E**). All data are represented as mean ± standard deviation (SD). * *p* < 0.05, ** *p* < 0.01 and *** *p* < 0.001 versus CUMS group.

**Figure 2 ijms-20-00195-f002:**
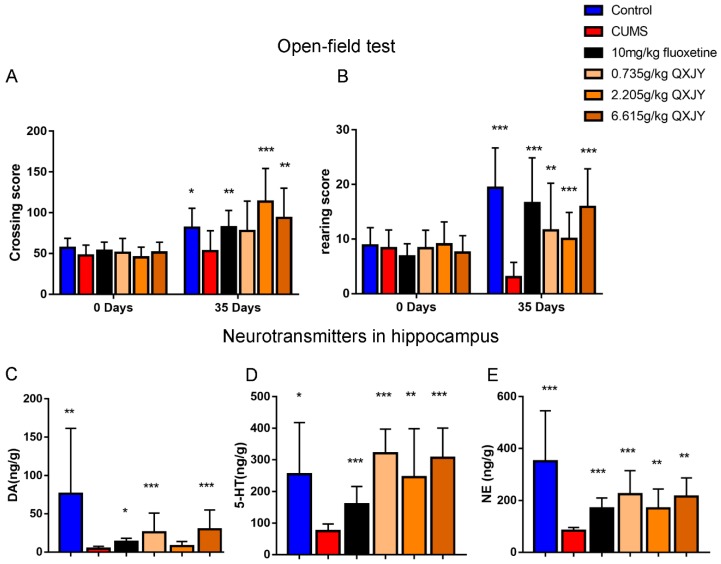
Effects of ZL on CUMS rats in OFT and neurotransmitters in hippocampus. Effects of ZL on OFT (**A**,**B**). Decrease of monoamine neurotransmitters in the hippocampus (**C**–**E**). All data are represented as mean ± SD. * *p* < 0.05, ** *p* < 0.01, and *** *p* < 0.001 versus CUMS group.

**Figure 3 ijms-20-00195-f003:**
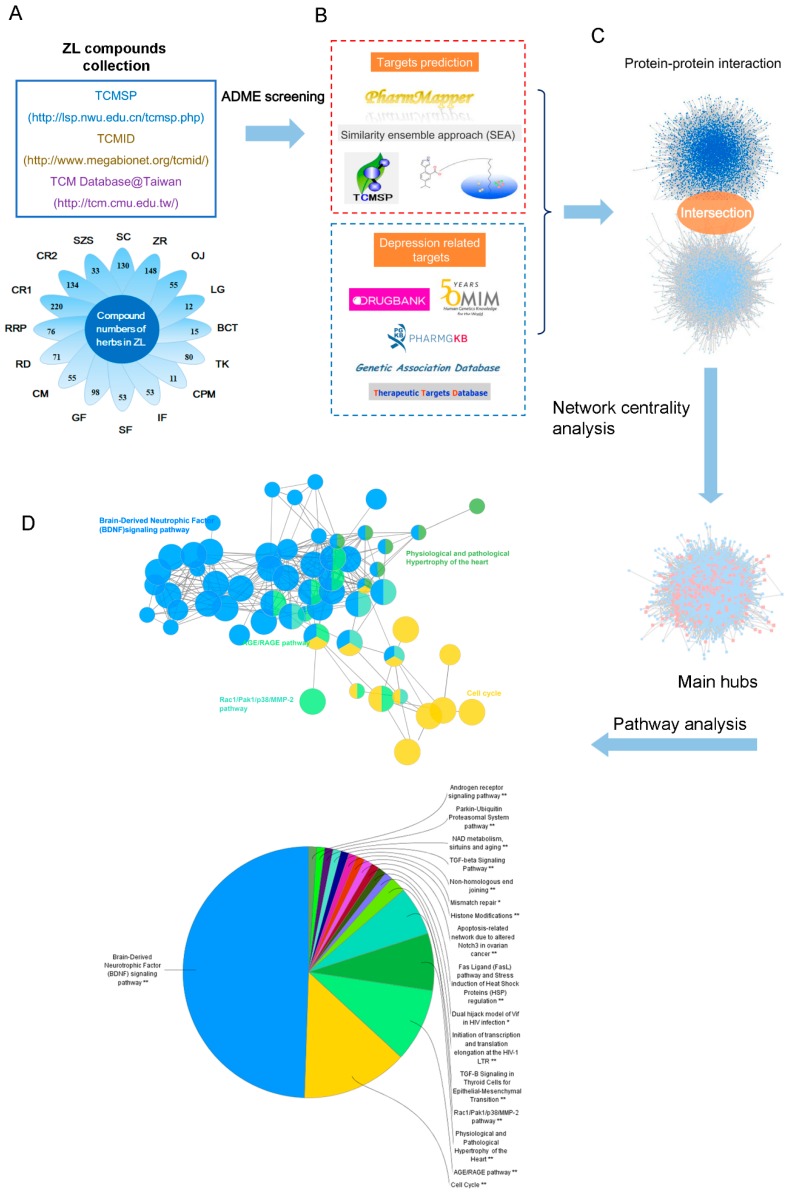
Systems pharmacology workflow. (**A**) Collection and screening of compounds from ZL; (**B**) active compound-target prediction and construction of known depression-related targets; (**C**) identification of therapeutic targets for ZL; (**D**) pathway enrichment analysis.

**Figure 4 ijms-20-00195-f004:**
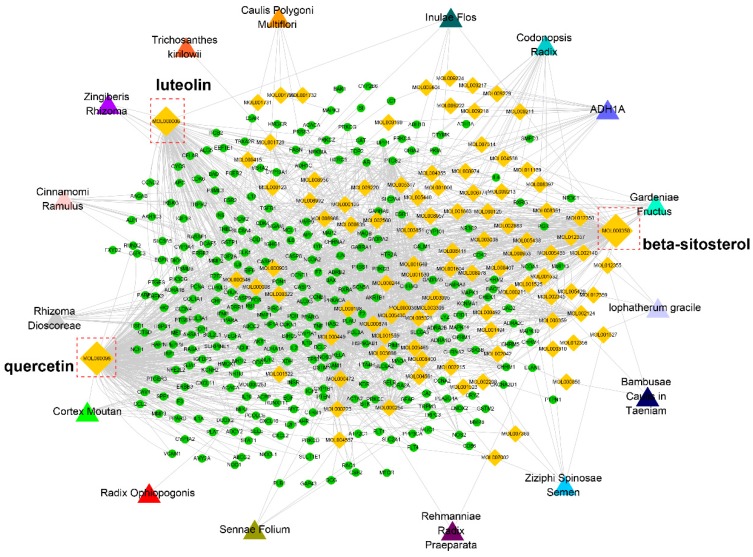
Compound-target network. The network was constructed by linking active compounds (marked by compound ID) from different herbs and their predicted targets.

**Figure 5 ijms-20-00195-f005:**
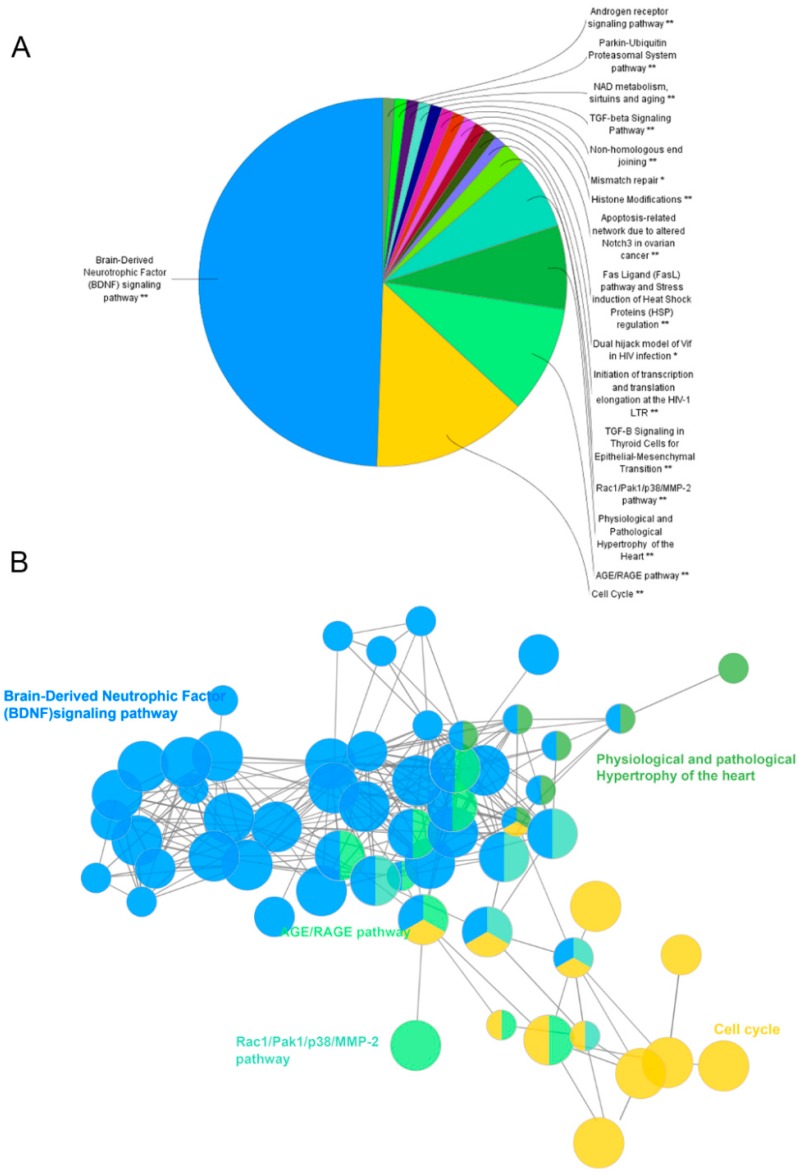
Pathway enrichment analysis. The analysis was conducted by ClueGo. GO terms are represented as nodes, and node size represents the term enrichment significance. Functionally related groups partly overlap. Representative enriched pathway (*p* < 0.05) interactions among candidate targets of ZL are shown. (**A**,**B**) Candidate ZL targets enriched in representative signaling pathways. Representative enriched pathways (* *p* < 0.05, ** *p* < 0.01) were shown.

**Figure 6 ijms-20-00195-f006:**
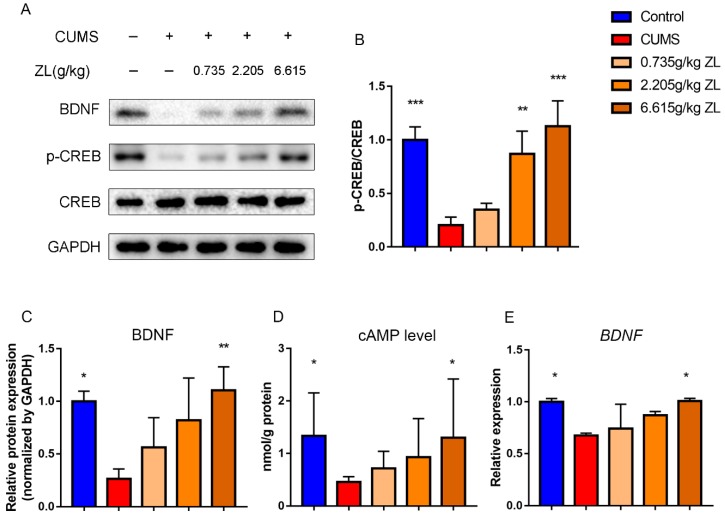
ZL activated cAMP-CREB-BDNF signaling. (**A**–**C**) Protein expressions of BDNF, p-CREB and CREB in the hippocampus at the end of experiment determined by western blotting. Quantified protein level was normalized to that of GAPDH using densitometry (*n* = 3). (**D**) cAMP levels in the rat hippocampus determined by ELISA (*n* = 10). (**E**) Relative gene expressions of BDNF in the rat hippocampus quantified by qPCR (*n* = 3). GAPDH was used as the loading control. All data are represented as mean ± SD. * *p* < 0.05, ** *p* < 0.01, and *** *p* < 0.001 versus CUMS group.

**Figure 7 ijms-20-00195-f007:**
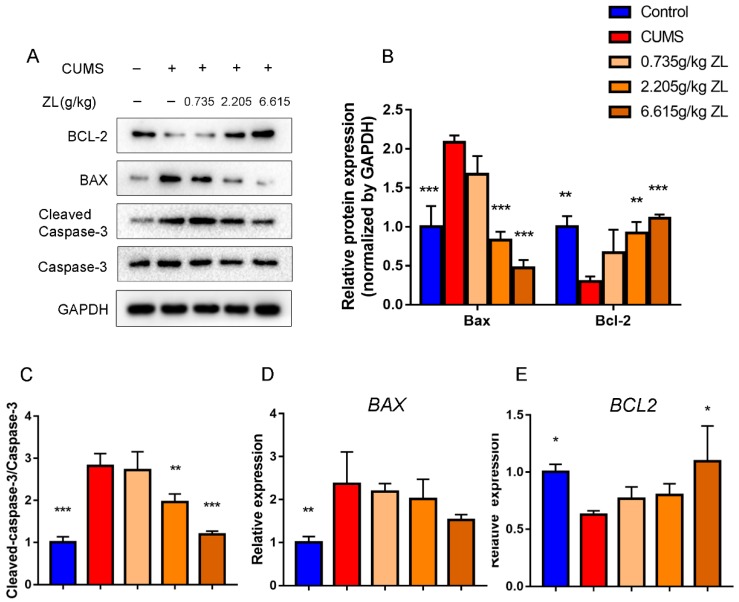
ZL exerted neuroprotective effects by inhibiting apoptosis. (**A**–**C**) Protein expressions of Bcl-2, Bax, cleaved caspase-3 and caspase-3 in the hippocampus at the end of experiment determined by western blotting. Quantified protein level was normalized to that of GAPDH using densitometry (*n* = 3). (**D**,**E**) Relative gene expressions of Bcl-2 and Bax in the rat hippocampus quantified by qPCR (*n* = 3). GAPDH was employed as the loading control. All data are represented as mean ± SD. * *p* < 0.05, ** *p* < 0.01, and *** *p* < 0.001 versus CUMS group.

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
