# Peer review of "Zhile Capsule Exerts Antidepressant-Like Effects through Upregulation of the BDNF Signaling Pathway and Neuroprotection"

_ijms, 2019, doi:10.3390/ijms20010195_

Reviewer 1 Report

As stressed by the authors, traditional Chinese medicine is a promising source for new treatments for depression. Zhile had beneficial effects in the rat model of chronic unpredictable mild stress. A systems pharmacology approach led to BDNF as a major depression relevant target of Zhile. Zhile had impressive dose-dependent rescue effects on cAMP-CREB-BDNF and apoptotic markers.

1) This is a well-done validation study utilizing a variety of methods. However, all the complicated multi-step systems pharmacology approach brought us back to the cAMP-CREB-BDNF signaling pathway known to be down-regulated in this model and in depression. May be worthwhile to give some thought to cell cycle and AGE/RAGE, the candidate ZL targets which came after BDNF. A potentially more exciting next step could be to use a discovery study to look for unknown targets affected by Zhile.

Abstract:

2) Undefined abbreviations (CUMS, ZL) in the abstract make it difficult to understand.

3) The concluding sentence, “It is beneficial to depressive patients in clinical practice” is unclear. If it is the starting point for this study, it should be at the beginning of the Abstract. If it is supposed to be the conclusion, it needs to be toned down to reflect what can be concluded from a rat model study.

Introduction:

4) What is Zhile used for in TCM? I could not find any papers on it in PubMed.

5) Are the abbreviations given to the components of Zhile used anywhere in the paper? If not, better to remove them.

Results:

6) The results do not refer to Fig 2A-B.

7) No need to repeat the list of the 16 components of Zhile, as it was already given in the Introduction.

8) May be easier for the readers to follow, if each step of the systems pharmacology workflow (3A, 3B, 3C, 3D) is referred to in the appropriate place in the text.

9) Line 118: OB, DL, Caco-2, BBB, HL should be spelled out here, even if a longer explanation is in the Materials and Methods (4.4.1).

10) Table S2: Please include which component(s) provided each target.

11) Fig 4: Too tiny to read. Highlight quercetin, beta-sitosterol and luteolin in the figure.

12) Line 161: the topological features should be spelled out here, even if explained in Materials and Methods (4.4.4).

13) Fig 5B: too small to be read, the names on pathways should be moved from top of figure to make it clearer.

14) Fig 6: Consider moving B next to A and D between C&E to make it more reader-friendly.

15) Minor:

Line 37: there seems to be a stray “48”

Line 43: There seems to be a stray “62”

Line 58: Although TCM was defined in Affiliations, it should be spelled out here.

Line 97: “ZF” should be “ZL”

Lines 154, 157: Spell out PPI and CPPI at first appearance.

Table S6: What is SUID? % Common Gene and %Genes Cluster # are empty, so useless.

Author Response

Response to Reviewer 1 Comments

We feel great thanks for your professional review on our article. As you are concerned, there are several problems that need to be addressed. According to your nice suggestions, we have made extensive corrections to our previous draft. The detailed corrections are listed below.

Point 1 : This is a well-done validation study utilizing a variety of methods. However, all the complicated multi-step systems pharmacology approach brought us back to the cAMP-CREB-BDNF signaling pathway known to be down-regulated in this model and in depression. May be worthwhile to give some thought to cell cycle and AGE/RAGE, the candidate ZL targets which came after BDNF. A potentially more exciting next step could be to use a discovery study to look for unknown targets affected by Zhile.

Response 1: Your advice is much greatly interesting. As a Traditional Chinese Medicine formula-based capsule, Zhile exert potent antidepressant effects in CUMS rat model. And due to the complexity of the Chinese medicine, we thus used a systems pharmacology approach to discover the possible mechanisms. Here, we only chose the most related pathway, cAMP-CREB-BDNF signaling pathway and did not pay too much attention to other enriched pathways such as cell cycle and RGE/RAGE. Your concern is valuable for our further study, and the authors will continue to focus on future research about unkown targets which affected by Zhile and we think this work may help us understand the pharmacological effects of Zhile better.

Point 2Undefined abbreviations (CUMS, ZL) in the abstract make it difficult to understand.

Response 2: Your advice is much appreciated and the authors have spelled out the abbreviations (CUMS, ZL) in the abstract so it has become much easier to understand.

Point 3: The concluding sentence, “It is beneficial to depressive patients in clinical practice” is unclear. If it is the starting point for this study, it should be at the beginning of the Abstract. If it is supposed to be the conclusion, it needs to be toned down to reflect what can be concluded from a rat model study.

Response 3: Your concern is much appreciated and we totally agreed with your suggestion. We have corrected the inappropriate sentence and made it clearer.

Point 4: What is Zhile used for in TCM? I could not find any papers on it in PubMed.

Response 4: The authors feel sorry about the fuzzy definition about Zhile. Zhile is a Traditional Chinese Medicine formula-based capsule which is composed of sixteen herbs, including Schisandra chinensis,  Semen Ziziphi Spinosae, Codonopsis Radix, Cinnamomi Ramulus, Rehmanniae Radix Praeparata, Rhizoma Dioscoreae, Cortex Moutan, Gardeniae Fructus, Sennae Folium, Inulae Flos, Caulis Polygoni Multiflori, Trichosanthes kirilowii, Bambusae Caulis in Taeniis, Lophatherum gracile, Ophiopogon japonicus and Zingiberis rhizome. Many drugs containing one of the above components have a variety of biological effects on the central nervous and immune systems, accompanied by antidepressant effects. For instance, Semen Ziziphi Spinosae alleviate depressive behaviors in depression models (PMID: 28454575); Moreover, Schisandra chinensis also have antidepressant effects (PMID: 28761074). However, TCM emphasizes compatibility and focuses on the integrated effects of all herbs in formula. Here, we evaluated the integrated antidepressant effects of the sixteen herbs in Zhile. 

Point 5: Are the abbreviations given to the components of Zhile used anywhere in the paper? If not, better to remove them.

Response 5: The abbreviations given to the components of Zhile did not used anywhere in the paper and in accordance with your request, we have already removed them.

Point 6: The results do not refer to Fig 2A-B.

Response 6: The authors feel sorry about such huge mistake and thanks for your concern. We have already add this part in the first result: 2.1. ZL attenuated depressive like behaviors of CUMS rats.

Point 7: No need to repeat the list of the 16 components of Zhile, as it was already given in the Introduction.

Response 7: Thanks for your suggestion and we have already removed them.

Point 8: May be easier for the readers to follow, if each step of the systems pharmacology workflow (3A, 3B, 3C, 3D) is referred to in the appropriate place in the text.

Response 8: Your advice is greatly valuable and we fully agreed this suggestion. Each step of the systems pharmacology workflow has been inserted into the main article when referred to the related procedure. It is now easier for readers to follow.

Point 9: Line 118: OB, DL, Caco-2, BBB, HL should be spelled out here, even if a longer explanation is in the Materials and Methods (4.4.1).

Response 9: Your advice is much appreciated and the authors have spelled out the abbreviations (OB, DL, Caco-2, BBB and HL).

Point 10: Table S2: Please include which component(s) provided each target.

Response 10: Your suggestion is much appreciated. We actually provided components and their related targets in Table S3. Table S2 is used to clarify the total targets Zhile involved in, so the authors did not add the related components. Readers may find more detailed Herb-Component-Target information in Supplementary Table S3.

Point 11: Fig 4: Too tiny to read. Highlight quercetin, beta-sitosterol and luteolin in the figure.

Response 11: Your concern is much appreciated. The authors noticed this problem and make some changes in Fig 4. Generally, we highlighted quercetin, beta-sitosterol and luteolin in the figure. We also move the elements so that readers will not see overlapping between triangles and circles. We also provided the 600 dpi resolution of Figure 4 and it may be easier to read now.

Point 12: Line 161: the topological features should be spelled out here, even if explained in Materials and Methods (4.4.4).

Response 12: Your advice is much appreciated and the authors have spelled out all the topological features.

Point 13: Fig 5B: too small to be read, the names on pathways should be moved from top of figure to make it clearer.

Response 13: We thank you very much for your suggestion. In accordance with your suggestion, we choose the Top 5 main pathways on Figure 5B and we deleted the remaining pathways. Figure 5A has already presented all the pathways for readers, so we present the main pathways and make Fig 5B much clearer. Thank you for your consideration.

Point 14: Fig 6: Consider moving B next to A and D between C&E to make it more reader-friendly.

Response 14: Thank you for your nice suggestion and we have already moved B next to A and D between C&E.

Point 15: Minor:

Line 37: there seems to be a stray “48”

Line 43: There seems to be a stray “62”

Line 58: Although TCM was defined in Affiliations, it should be spelled out here.

Line 97: “ZF” should be “ZL”

Lines 154, 157: Spell out PPI and CPPI at first appearance.

Table S6: What is SUID? % Common Gene and %Genes Cluster # are empty, so useless.

Response 15: We greatly thank you for your comments for pointing out our small mistakes here and as the reviewer suggested, we deleted the Line 37: stray “48” and Line 43: stray “62”. We also spelled out Traditional Chinese Medicine (TCM) in the manuscript. Moreover, we corrected Line 97: ZF and spell out Protein-protein interaction (PPI) and core protein-protein interaction (CPPI) at its first appearance. We have deleted the useless lists in Supplementary Table S6. Unfortunately, we did not find out clearly what is SUID. But we found out SUID is related to the plugin itself. When we put the same genes into ClueGO plugin and we obtained the same results in pathways and any other results but only the different SUID. We also found that in the same session, the SUID become larger when we conducted new ClueGO analysis using the same genes without affecting other results, and when we started a new session using different genes and we still obtained the same SUID but different pathways. In general, SUID may be related to the analysis numbers in one ClueGO session and did not related to the specific pathway. So we deleted SUID in case readers will have the same confusion.

Finally, the authors tried our best to improve the manuscript and made some changes in the manuscript.

We appreciate for Editors/Reviewers’ warm work earnestly, and hope that the correction will meet with approval.

Once again, thank you very much for your comments and suggestions.

Sincerely yours,

Aiyun Wang, Yin Lu

Reviewer 2 Report

Results of this interesting work are somewhere not clear in that language and style seem to be too much technical and less comparaison toward available antidepressants agent is given. Could you please improve the work in this direction from results right to conclusion? 

Author Response

Response to Reviewer 2 Comments

Thank you for your nice comments on our article. We really appreciate your help and patience. Based on your comments, we attached an answer letter to you. We have made extensive revisions to our previous draft. The detailed point-by-point responses are listed below.

Point 1: Results of this interesting work are somewhere not clear in that language and style seem to be too much technical and less comparaison toward available antidepressants agent is given. Could you please improve the work in this direction from results right to conclusion?

Response 1: Thank you for your comments and we totally agree with your suggestions which might be of great help to improve the quality of our manuscript. We feel sorry about the language and style in results, and we revised our results in language seriously. We try to make all results more connectively and easier for readers to follow.

In accordance to your second concern: “less comparaison toward available antidepressants agent is given”, during the pharmacodynamic evaluation of Zhile Capsule on depression, we had already used fluoxetine hydrochloride (Lilly suzhou pharmaceutical co., LTD, Cat No. 6620A) as a positive control drug to assess whether our chronic unpredictable mild stress model is successful. Thank you for your suggestion and we totally agree your comment. We have already rearranged the experiment data and we also compared the antidepressant-like effects between fluoxetine and Zhile Capsule in our new manuscript. Your advice is very helpful for us to evaluate the antidepressant-like effects of Zhile. Thanks for your suggestions again.

Finally, the authors tried our best to improve the manuscript and made changes in the manuscript. We added 10mg/kg fluoxetine group in Fig 1 and Fig 2 and change the related figures. We also made some changes which may not influence the content and framework of the paper using "Track Changes" function in Microsoft Word.

We appreciate for Editors/Reviewers’ warm work earnestly, and hope that the correction will meet with approval.

Once again, thank you very much for your comments and suggestions.

Sincerely yours,

Aiyun Wang, Yin Lu